# DAU-Net: Dual attention-aided U-Net for segmenting tumor in breast ultrasound images

**Payel Pramanik[1☉], Ayush Roy[2☉], Erik Cuevas[3], Marco Perez-Cisneros[4]\*, Ram Sarkar[1]**

**1** Department of Computer Science and Engineering, Jadavpur University, Kolkata, India, **2** Department of Electrical Engineering, Jadavpur University, Kolkata, India, **3** Departamento de Electrónica, Universidad de Guadalajara, Guadalajara, Mexico, **4** División de Tecnologías Para La Integración Ciber-Humana, Universidad de Guadalajara, Guadalajara, Mexico

☉ These authors contributed equally to this work.
\* marco.perez@cucei.udg.mx

**Data Availability Statement:** The publicly available dataset is analyzed in this study. The data can be found here: [URL: 1. BUSI: https://www.kaggle.com/datasets/aryashah2k/breast-ultrasound-images-dataset (accessed on 20th February, 2023)

## Abstract

Breast cancer remains a critical global concern, underscoring the urgent need for early detection and accurate diagnosis to improve survival rates among women. Recent developments in deep learning have shown promising potential for computer-aided detection (CAD) systems to address this challenge. In this study, a novel segmentation method based on deep learning is designed to detect tumors in breast ultrasound images. Our proposed approach combines two powerful attention mechanisms: the novel Positional Convolutional Block Attention Module (PCBAM) and Shifted Window Attention (SWA), integrated into a Residual U-Net model. The PCBAM enhances the Convolutional Block Attention Module (CBAM) by incorporating the Positional Attention Module (PAM), thereby improving the contextual information captured by CBAM and enhancing the model's ability to capture spatial relationships within local features. Additionally, we employ SWA within the bottleneck layer of the Residual U-Net to further enhance the model's performance. To evaluate our approach, we perform experiments using two widely used datasets of breast ultrasound images and the obtained results demonstrate its capability in accurately detecting tumors. Our approach achieves state-of-the-art performance with dice score of 74.23% and 78.58% on BUSI and UDIAT datasets, respectively in segmenting the breast tumor region, showcasing its potential to help with precise tumor detection. By leveraging the power of deep learning and integrating innovative attention mechanisms, our study contributes to the ongoing efforts to improve breast cancer detection and ultimately enhance women's survival rates. The source code of our work can be found here: https://github.com/AyushRoy2001/DAUNet.

## Introduction

As per the World Health Organization (WHO), breast cancer stands as the most frequently diagnosed cancer and is the primary cause of cancer-related fatalities in women globally. In

2. UDIAT: http://www2.docm.mmu.ac.uk/STAFF/M. Yap/dataset.php.

**Funding:** The author(s) received no specific funding for this work.

**Competing interests:** The authors have declared that no competing interests exist.

the year 2020, approximately 2.3 million new instances of breast cancer were detected, constituting around 11.7% of all cancer cases worldwide. It also caused approximately 685,000 deaths, representing 6.9% of all cancer-related deaths globally [1]. Therefore, early and accurate detection of breast cancer is essential for improving treatment outcomes and patient survival rates. Among various medical imaging techniques, ultrasound imaging has played an important role in the detection and diagnosis owing to its non-invasive nature, and capability to capture high-resolution images of breast tissue [2]. However, the precise segmentation of breast lesions in ultrasound images remains a challenging task because of the presence of speckle noise, poor image quality, and inherent variations in breast tissue [3]. Manual segmentation of ultrasound images is a time-consuming task. As a result, the implementation of automatic segmentation techniques becomes important to enhance efficiency and minimize unnecessary delays [4]. Sample breast ultrasound images are shown in Fig 1.

In recent times due to the advancements in deep learning techniques, medical image analysis and CAD systems have undergone a revolutionary transformation. There are many CAD systems that aim to automate the process of breast lesion segmentation in ultrasound images, assisting radiologists in accurate diagnosis. For instance, the popularity of Convolutional Neural Networks (CNNs) has increased for automatic detection and segmentation of breast cancer in ultrasound images. These models can learn feature representations from raw pixel intensities, enabling them to capture complex patterns and discriminative features [5]. Further, to extract the contextual information, authors of [6] introduced the U-Net model. The basic U-Net model utilizes an encoder-decoder structure, with the encoder capturing context from the input image, and the decoder generating a segmentation map through upsampling. This architecture includes skip connections between encoder and decoder layers to preserve spatial information at multiple scales. Since its introduction, U-Net has served as a foundation for several variants and extensions. Variants of U-Net include the Residual U-Net [7], which incorporates residual blocks to facilitate gradient flow and address vanishing gradient issues, the Attention U-Net [8], which integrates attention mechanisms to selectively focus on informative regions, the U-Net++ [9], which introduces nested skip connections to capture more comprehensive contextual information, the Hybrid U-Net, which combines U-Net with other architectures like VGG or ResNet for improved performance and many more [10]. These variants of U-Net have demonstrated advancements in image segmentation tasks. In a variety of computer vision tasks, attention mechanisms have produced promising results. The Attention U-Net, for instance, introduces attention gates to emphasize relevant features and suppress irrelevant ones. Other attention mechanisms, such as Squeeze-and-Excitation (SE), Channel-Attention Mechanism (CAM) and Spatial-Attention Mechanism (SAM) have also been applied to improve feature representations in CNNs [11, 12]. While U-Net and its variants excel in capturing local context, integrating global contextual information has been explored as a means to improve segmentation accuracy. Studies such as [13–16] have introduced different models to capture global contextual information effectively.

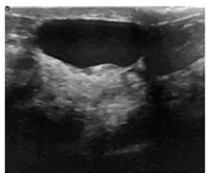 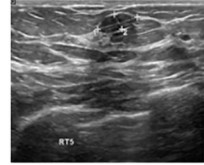 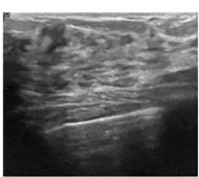

Benign ultrasound Malignant ultrasound Normal ultrasound

**Fig 1. Sample breast ultrasound images of benign, malignant, and normal types.**

## Contributions

In light of the existing literature, we propose a novel segmentation method for detecting tumors in breast ultrasound images. Our approach applies two attention mechanisms, namely the Positional Convolutional Block Attention Module (PCBAM) and the Shifted Window Attention (SWA), which effectively capture context-aware features, spatial relationships, and global contextual information. The entire architecture of the proposed segmentation model is shown in Fig 2.

The highlights of this work are as follows:

- Our proposed model uses two attention mechanisms in the U-Net model, called PCBAM and SWA.

- PCBAM, the Position attention aided CBAM, combines the CBAM attention mechanism, which captures context-aware features through channel attention and spatial attention, with positional attention. This integration enhances the contextual information and spatial relationships within local features, leading to more robust and accurate representations.

- SWA is used in the bottleneck layer of the Residual U-Net model to capture global contextual information.

- The combination of PCBAM and SWA significantly improves the performance of the model on both the BUSI and UDIAT datasets.

The paper is organized as follows. First, we review the segmentation methods used in breast cancer ultrasound imaging by various researchers and identify the existing gaps in the literature. Next, we discuss the preliminary details and the proposed model for tumor region segmentation in breast ultrasound images. The evaluation of the model using various metrics and the analysis of the results are discussed then. Finally, we conclude our work with some potential future research directions.

## Related work

Breast cancer is a pressing global health issue, and hence researchers have been exploring various methods to improve early detection, and precise diagnosis to enhance survival rates for

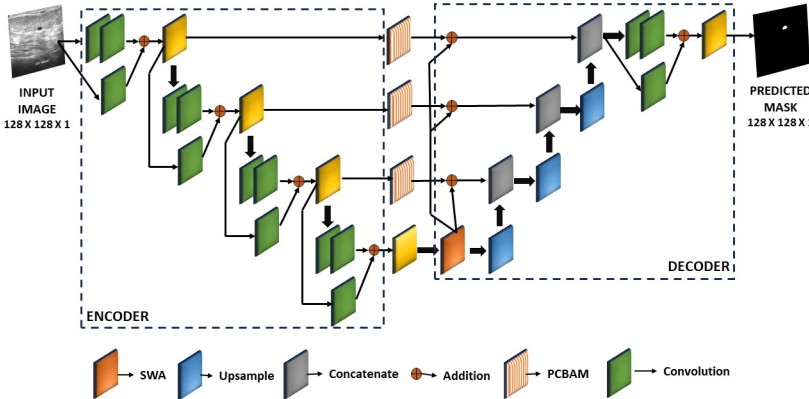

**Fig 2. Block diagram of the proposed DAU-Net model used for segmentation of tumor in breast ultrasound images.** An input image with dimensions 128 × 128 × 1 undergoes feature extraction through the encoder, and the decoder then performs upsampling on the encoded features to predict a binary mask of size 128 × 128 × 1. The in-between connections of the encoder and the decoder are accompanied by the addition of PCBAM and SWA attention mechanisms to enhance the performance.

affected women [17–20]. Among these efforts, deep learning-based CAD systems have shown great promise to address this challenge. This section presents a review of relevant literature that centers around deep learning-based segmentation methods for detecting tumor regions in breast ultrasound images. Deep learning methods, especially CNNs, have shown impressive achievements in diverse medical imaging tasks, such as image segmentation, classification, and detection [2, 21–23]. Researchers have applied CNNs to analyze breast ultrasound images to detect abnormalities and tumors [24–26]. Studies such as [27–30] explored different architectures and attention mechanisms to improve the performance of tumor segmentation in breast ultrasound images. In the study by Vakanski et al. [27], the authors combined visual saliency into a U-Net model. By incorporating visual saliency maps that capture regions attracting radiologists' attention and combining topological and anatomical prior knowledge, the model learned feature representations prioritizing essential spatial regions. However, a limitation of this approach lies in its reliance on the quality of saliency maps, as using low-quality maps may not enhance results and could potentially lead to degraded performance. In another study by Lee et al. [30], the authors proposed a semantic segmentation network to enhance the accurate segmentation of regions of breast tumors in ultrasound images. They achieved this improvement by integrating a channel attention module with multi-scale grid average pooling (MSGRAP). This attention module enables the utilization of both global and local spatial information from input images, thereby enhancing the network's effectiveness in performing semantic segmentation. Chen et al. introduced AAU-Net [31], which is a hybrid adaptive module, combining convolutional layers with varying kernel sizes, channel self-attention, and spatial self-attention blocks to replace the traditional convolution operation. In contrast, in another study [32], the authors introduce a cascaded CNN, which integrates U-Net, Bidirectional Attention Guidance Network (BAGNet), and Refinement Residual Network (RFNet). CBAM introduced by Woo et al. [33] demonstrated significant potential in improving the capability of CNNs to focus on relevant image regions. By integrating both channel and spatial attention mechanisms, CBAM enhances the representational power of CNNs and boosts performance in various computer vision tasks [34]. Researchers have utilized CBAM attention in tumor detection from breast cancer imaging [35–37]. In [37], the introduced method employed a deep ResNet architecture with a CBAM attention module to extract more comprehensive and in-depth features from pathological images. In [35], the authors introduced a semi-supervised learning model named BUS-GAN, comprising two networks: BUS-S for segmentation and BUS-E for evaluation. The BUS-S network extracts features of multi-scale to handle variations in breast lesions, enhancing segmentation robustness. To enhance discriminative ability, the BUS-E network incorporates a dual-attentive-fusion block with spatial attention paths, distilling geometrical and intensity-level information from both the segmentation map and the original image. Through adversarial training, the BUS-GAN model achieves higher segmentation quality as the BUS-E network guides BUS-S in generating more precise segmentation maps that align closely with the ground truth distributions. Another study by Fan et al. [36] showed an approach called the Multi-Task Learning (MTL) approach to address joint breast tumor lesion localization and classification. The model comprises a classifier, an auxiliary lesion-aware network, and a shared feature extractor. Multiple attention modules are incorporated in the auxiliary network to optimize the multi-scale intermediate feature maps, and enhance representativeness through channel and spatial attention focused on lesion regions. Positional attention mechanisms have gained attention in the medical imaging domain due to their ability to capture spatial relationships and contextual information within local features. The authors of [38] utilized positional attention in a multi-scale framework to identify anatomical structures in medical images. SWA, a recent innovation proposed by [39], enhances the efficiency and adaptability of the attention mechanism. By applying the attention

mechanism in a sliding window fashion, SWA effectively captures relevant information across multiple scales, making it suitable for object detection and segmentation tasks. On the other hand [40], investigated the effectiveness of an ensemble of Swin transformers for two-class (benign vs. malignant) and eight-class (four benign and four malignant sub-types) classification in medical imaging, using the BreaKHis histopathology dataset. Swin transformer is a variant of vision transformer that utilizes non-overlapping SWA. Another approach, presented by [41], introduced the BTS-ST network, which combines Swin-transformer with CNN-based U-Net for breast tumor segmentation and classification. The BTS-ST network incorporates SWA to enhance feature representation capability for irregularly shaped tumors. The residual U-Net architecture, introduced in [7], is a variant of the traditional U-Net model. Incorporating residual connections allows for better information flow during training and helps mitigate the vanishing gradient problem, leading to improved convergence and performance. Several studies have explored breast tumor segmentation using residual U-Net-based deep-learning techniques. For instance, the authors of [42] presented the RCA-IUnet, a deep-learning model designed for breast tumor segmentation in ultrasound imaging. The model integrates U-Net architecture with residual inception depth-wise separable convolution, hybrid pooling, and cross-spatial attention filters in long skip connections, effectively extracting tumor-related features. The authors conducted an ablation study, highlighting the pivotal role of residual inception convolution and cross-spatial attention components in the proposed model. However, a limitation of the model is the absence of a channel attention filter, which may restrict its capacity to emphasize the most critical feature layers. In another work reported in [43], an improved U-net MALF model was proposed for breast tumor segmentation in ultrasound images. This model enhances the attention U-net network framework by incorporating residual convolution and extended residual convolution modules in the encoding path. In their work [44], utilized a residual U-Net for breast tumor segmentation, incorporating a fusion attention mechanism that combines both spatial and channel attention. In another work [45], the authors presented the RDAU-NET (Residual-Dilated-Attention-Gate-UNet) model for tumor segmentation in breast ultrasound images. The model extends the conventional U-Net architecture and includes three modules: Residual unit, Dilation unit, and Attention Gate. These modules are introduced to improve the model's performance and capabilities for accurate segmentation of breast tumors in ultrasound images. Several comparative studies have evaluated different deep learning-based models along with attention mechanisms for breast tumor segmentation in ultrasound images. The authors of [27, 46, 47] compared the performance of various CNN architectures with attention mechanisms, showing the potential of attention-based methods in improving segmentation accuracy. In summary, while deep learning-based methods have shown promise in breast tumor segmentation, there is still a need to explore more advanced attention mechanisms further to improve the accuracy and robustness of the models.

## Methodology

Our research showcases a novel methodology that integrates PCBAM and SWA with the Residual U-Net design. This design consists of two crucial elements, the encoder and decoder, that collaborate to extract significant attributes from input images and produce accurate segmented results.

### Encoder

To extract hierarchical features from the input data, the encoder employs convolutional layers with $3 \times 3$ filters and a stride of 1. Batch normalization and ReLU activation are applied after

each convolutional layer to maintain feature stability and increase information flow. To ensure smooth gradient flow during training and to retain essential information, residual connections are utilized. To downsample, $2 \times 2$ stride convolutional layers are employed. Additionally, the PCBAM mechanism improves the encoder features before connecting them with the decoder features via residual connections. More information on this mechanism can be found in the subsequent Section.

## Decoder

In the process of upsampling and reconstructing segmented output, the decoder plays a crucial role. By combining upsampled feature maps with those from the attention-aided encoder features, it gains access to both low-level and high-level features. This happens through strategic fusion, which involves refining the features with convolutional layers, followed by batch normalization and ReLU activation. As a result, a higher-dimensional representation of the spatial relationships is obtained. To further enhance the spatial dimensions, the decoder uses residual blocks, which contribute to its exceptional performance. Moreover, the SWA layer is incorporated in the decoder, capturing global dependencies and improving spatial coherence in the segmentation results.

## Positional convolutional block attention module

CBAM attention mechanism [33] is applied to the last feature map of dimension $C \times H \times W$ generated from any CNN architecture. Here, $C$, $H$ and $W$ represent a feature map's number of channels, height, and width, respectively. The CBAM attention mechanism consists of two components: the 1D Channel Attention Module (CAM) and the 2D Spatial Attention Module (SAM). The CAM assigns weights to the channels of the feature map, enhancing specific channels that contribute more to improve model performance. It is formulated as per Eq 1.

$$F_c = \sigma(mlp(gap(F)) + mlp(gmp(F))) \tag{1}$$

In Eq 1, $\sigma$ represents the sigmoid activation function, $gap$ is the global average pooling layer, $gmp$ is the global max pooling layer, and $mlp$ denotes the multi-layer perceptron consisting of two successive fully connected i.e., dense layers (DL) with $C$ and $C/8$ units, respectively and $F$ is the feature map. Now, $F'_c = F_c \otimes F$ is fed to the SAM ($\otimes$ denotes the element-wise matrix multiplication).

The SAM operates on the feature map $F'_c$ obtained from the CAM. It applies a spatial attention mask to enhance the feature representation. The SAM is formulated according to Eq 2.

$$F'' = f^{7 \times 7}[DL(gap(F'_c)); DL(gmp(F'_c))] \tag{2}$$

In Eq 2, $f^{7 \times 7}$ is a convolutional layer with a kernel size of $7 \times 7$ and dilation of 4, $DL$ represents the dense layers, and ';' denotes the concatenation operation. The final output feature map of the CBAM attention module, denoted as $F_{CBAM}$, is obtained by element-wise multiplication between $F''$ and $F'_c$ as shown in Eq 3.

$$F_{CBAM} = F'' \otimes F'_c \tag{3}$$

The CBAM attention mechanism effectively captures channel-wise and spatial-wise dependencies, thereby allowing the model to focus on relevant features, and improve its performance in image segmentation.

Similarly, the Position Attention Module (PAM) is designed to enrich local features by incorporating a broader context, thereby enhancing their representational capacity. To achieve

this, we start with a local feature map denoted as $F \in \mathbb{R}^{H \times W \times C}$. This feature map is processed through a convolutional layer, resulting in two new feature maps, $B$ and $Z$, both of size $R^{H \times W \times C}$. Afterward, $B$ and $Z$ are reshaped into matrices of size $\mathbb{R}^{N \times C}$, where $N = H \times W$, representing the number of pixels in the feature map. A matrix multiplication is performed between the transpose of $Z$ and $B$, followed by the application of a softmax layer, which yields the spatial attention map $S \in R^{N \times N}$. This attention map captures the spatial relationships between different pixels in the feature map. PAM allows local features to leverage a wider contextual understanding by employing the attention mechanism to emphasize relevant spatial information. This enables the local features to better represent complex patterns and structures in the input data. The formula is shown in Eq 4.

$$s_{ji} = \frac{\exp(B_i \cdot Z_j)}{\sum_{i=1}^{N} \exp(B_i \cdot Z_j)} \tag{4}$$

where $s_{ji}$ measures the impact of the $i$th position on the $j$th position.

Next, we feed feature map $F$ into a convolutional layer to generate a new feature map $D \in R^{H \times W \times C}$, which is reshaped to $R^{N \times C}$. We perform a matrix multiplication between $D$ and the transpose of $S$, resulting in a feature map of size $R^{N \times C}$. We then reshape this back to $R^{H \times W \times C}$. Finally, we multiply it by a scale parameter $\alpha$ and perform an element-wise sum operation with the features $F$ to obtain $F_{PAM} \in R^{H \times W \times C}$. The calculation is done in accordance with Eq 5.

$$F_{PAM_j} = \alpha \sum_{i=1}^{N} (s_{ji} \cdot D_i) + F_j \tag{5}$$

where $\alpha$ is initialized as 0. The model learns $\alpha$ and gradually learns to assign more weight. The resulting feature $F_{PAM}$ at each position is a weighted sum of the features across all positions and original features, allowing for a global contextual view and selective aggregation of contexts based on the spatial attention map. This promotes intra-class compactness along with semantic consistency within the feature representations.

Utilizing the power of CBAM and PAM, we combine these two modules using Eq 6 to formulate the PCBAM, where the input feature to both CBAM and PAM is $F$. The block diagram of the PCBAM is shown in Fig 3.

$$F_{PCBAM} = F_{PAM} + F_{CBAM} \tag{6}$$

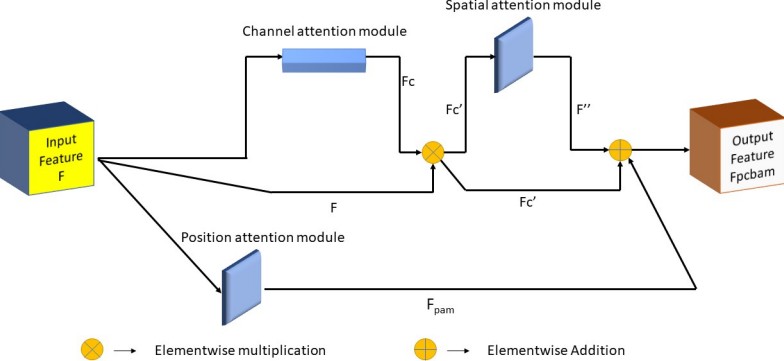

**Fig 3. An illustration of the PCBAM attention block.** CBAM and PAM are applied to the input feature F. The addition of the outputs of CBAM and PAM is the output of the PCBAM attention mechanism, $F_{PCBAM}$.

## Shifted window attention

The SWA [39] is a powerful attention mechanism used to capture global dependencies and improve spatial coherence in the segmentation results of our proposed model. It enhances the model's ability to focus on relevant regions and strengthens its contextual understanding of the input images. In image segmentation tasks, understanding the contextual relationships among different regions is crucial. However, traditional convolutional operations might not fully capture these long-range dependencies. To this end, we use the SWA module in order to address this limitation by introducing a window-based attention mechanism, which allows the model to attend to relevant information from different parts of the image.

The SWA mechanism can be mathematically defined as follows. Let $F$ be the input feature map of size $H \times W \times C$, where $H$, $W$, and $C$ represent the height, width, and number of channels, respectively. To compute the attention map, we first obtain position-aware query matrix $q$, key matrix $k$, and value matrix $v$ as follows:

$$q_{i,j} = F_{i,j} \cdot w_q \tag{7}$$

$$k_{i,j} = F_{i,j} \cdot w_k \tag{8}$$

$$v_{i,j} = F_{i,j} \cdot w_v \tag{9}$$

where $w_q$, $w_k$, and $w_v$ are learnable weight matrices for query, key, and value projections, respectively.

Next, we perform a convolution operation $f^{1\times1}$ ($1 \times 1$ is the kernel dimension) on $q$, $k$, and $v$ to compute the attention map $A$ as per Eq 10.

$$A = f^{1\times1}(q, k, v) \tag{10}$$

The attention map $A$ is then added element-wise to the original feature map $F$ using a residual connection to obtain the final output of the SWA mechanism $X_{out}$ using the following Eq 11.

$$X_{out} = F + A \tag{11}$$

The SWA mechanism is integrated into the decoder part of the Residual U-Net architecture. By introducing SWA, the model can effectively capture long-range dependencies and achieve better spatial coherence in the segmentation results, leading to improved performance in segmenting breast tumor regions in ultrasound images.

## Loss function

The Dice loss [48], Binary Cross Entropy (BCE) [49] loss, and Focal loss [50] are popular loss functions in image segmentation tasks. These loss functions help guide the training process of segmentation models by quantifying the similarity between the ground truth and the predicted masks.

The Dice loss is derived from the Dice Coefficient, also known as the F1 Score. It measures the overlap between the ground truth and the predicted masks, aiming to maximize their similarity. The Dice loss is computed using Eq 12.

$$Dice\ Loss = 1 - \frac{2 \times TP}{2 \times TP + FP + FN} \tag{12}$$

In Eq 12, TP represents the number of true positives (correctly identified foreground)

pixels, FP represents the number of false positives (incorrectly identified foreground) pixels, and FN represents the number of false negatives (missed foreground) pixels.

The BCE loss is another widely used loss function for image segmentation. It measures the dissimilarity between the predicted and ground truth masks, aiming to minimize their difference. The BCE loss is computed using the following Eq 13.

$$BCE\ Loss = -\frac{1}{N}\sum_{i=1}^{N}(y_i \log(p_i) + (1 - y_i)\log(1 - p_i)) \tag{13}$$

In Eq 13, $N$ represents the total number of pixels, $y_i$ represents the ground truth label (foreground or background) for pixel $i$, and $p_i$ represents the predicted probability of the foreground class for pixel $i$.

The Focal loss is designed to address class imbalance in segmentation tasks and provide more focus on hard-to-classify pixels. It assigns higher weights to misclassified pixels and thus reduces the impact of easy-to-classify pixels during training. The Focal loss is computed using the following Eq 14.

$$Focal\ Loss = -\frac{1}{N}\sum_{i=1}^{N}(\alpha(1 - p_i)^{\gamma}\log(p_i)) \tag{14}$$

In Eq 14, $\alpha$ is a balancing parameter to control the contribution of each class, and $\gamma$ is the focusing parameter that modulates the rate at which the loss focuses on hard-to-classify pixels.

In our model training, we have used a combination of the Dice loss, BCE loss, and Focal loss as shown in Eq 15 to guide the optimization process. By minimizing this combined loss during training, our model learns to accurately segment the desired regions of interest.

$$Loss = Dice\ Loss + BCE\ Loss + Focal\ Loss \tag{15}$$

## Statement of ethical approval

All procedures performed in studies involving human participants were in accordance with the ethical standards of the institutional and/or national research committee and with the 1964 Helsinki Declaration and its later amendments.

## Results and discussion

### Dataset description

The BUSI dataset [51] includes ultrasound images from 600 female patients aged 25 to 75 years, collected at Baheya Hospital in Cairo, Egypt. The dataset contains 437 benign cases, 210 malignant cases, and 133 images of normal breast tissue, comprising a total of 730 diverse breast ultrasound images for research purposes. The images are in PNG format, and on average, it has a size of 500 × 500 pixels. In our research, we have utilized various benign and malignant cases, along with their corresponding masks, to both train and test our segmentation model. These masks are crucial in identifying areas of interest in ultrasound images and serve as ground truth annotations. Our evaluation process involves comparing the model's predictions with the actual target regions to gauge its performance.

### Evaluation metrics

The performance of our segmentation model is evaluated using commonly used metrics: Dice score, Intersection over Union (IoU) score, accuracy, recall, and precision. These

metrics provide quantitative measures of the model's ability to accurately delineate regions of interest.

**Accuracy.** The accuracy metric assesses the overall correctness of binary segmentation and is calculated as the ratio of correctly classified pixels to the total number of pixels. It is defined in Eq 16.

$$Accuracy = \frac{\text{TP + TN}}{\text{TP + TN + FP + FN}} \tag{16}$$

**Precision.** Precision evaluates the fraction of true positive predictions among all positive predictions and is defined in Eq 17.

$$Precision = \frac{\text{TP}}{\text{TP + FP}} \tag{17}$$

**Recall.** Recall, commonly referred to as sensitivity or true positive rate, quantifies the proportion of true positive predictions out of all the actual positive instances and is defined in Eq 18.

$$Recall = \frac{\text{TP}}{\text{TP + FN}} \tag{18}$$

**Intersection over Union.** IOU is a measure that quantifies the overlap between the ground truth mask and the predicted binary segmentation mask. It is calculated as the ratio of the intersection area between the two masks to their union area and is defined in Eq 19.

$$IoU = \frac{\text{TP}}{\text{TP + FP + FN}} \tag{19}$$

**Dice score.** The Dice score, also referred to as the F1 score, integrates both precision and recall into a single value for evaluation and is defined in Eq 20.

$$Dice\ score = \frac{2 \times \text{TP}}{2 \times \text{TP + FP + FN}} \tag{20}$$

## Experimental setup

We have developed our segmentation model using Python and have leveraged the TensorFlow and Keras libraries for implementation. For data manipulation and preprocessing, we have utilized numpy, OpenCV, and scikit-learn libraries, which have facilitated the efficient handling of data. To speed up training and utilize hardware acceleration, we use the high-performance NVIDIA TESLA P100 GPU.

## Hyperparameter details

Our model is trained for 50 epochs, where every epoch represents one complete pass through the entire dataset. To address the issue of non-uniform sizes in the original BUSI images, we resize all images to a uniform size of 128 × 128 pixels, which are input into the model for segmentation. In the architecture's convolutional layers, we utilize the 'He Normal' weight

initialization, which has proven to be effective in deep neural network architectures. This initialization strategy contributes to better convergence and performance during training. During the training phase, we use the Adam optimizer with a learning rate of 0.0001 to optimize the model. This choice of optimizer allows us to efficiently update the model's parameters, enhancing convergence during training. To ensure a comprehensive assessment, we have divided the data into a 70-10-20% train-test-validation split. The model is trained using 70% of the data, while the remaining subsets are reserved for testing and validation purposes. We have leveraged the training set to optimize the model's parameters, fine-tune hyperparameters using the validation set, and gauge the model's ability to generalize to new data by utilizing the testing set.

## Ablation study

A series of experiments are conducted to refine our segmentation model and evaluate the impact of various modifications. These experiments include:

*(i)* Base Residual U-Net model, serving as the initial benchmark.

*(ii)* Residual U-Net model with PAM applied to the skip connections.

*(iii)* Residual U-Net model with CBAM applied to the skip connections.

*(iv)* Residual U-Net model with PCBAM, combining the strengths of PAM and CBAM.

*(v)* Proposed model with PCBAM and SWA, emphasizing global features.

Results in Table 1 showcase the efficacy of each modification. Each model has been trained using the linear combination of Dice, BCE, and Focal loss. The addition of PAM and CBAM improves performance, while SWA further enhances accuracy and segmentation quality.

Fig 4 visually illustrates the performance enhancement achieved with attention mechanisms. The combination of PCBAM and SWA results in improved performance for both the small and large region of interest, refining feature representations and capturing both global and local spatial dependencies for accurate segmentation.

Through these experiments and analyses, we are able to improve our segmentation model iteratively, identifying the most effective modifications and attention mechanisms. These advancements make a significant contribution to enhancing the accuracy and robustness of our model, positioning it as an advanced solution for segmenting breast tumors in ultrasound images.Additionally, we experiment with a five-fold cross validation [52] approach for assessing the model's generalizability, and tabulate the results under Table 2.

## Statistical analysis

We have conducted a statistical test to assess the robustness of the proposed segmentation model compared to the other models considered in the ablation study. We hypothesize that

**Table 1. Performance metrics of the segmentation models.** All values are in %. Bold values indicate superior performance. The results are in $x(\pm y)$ format, where x is the mean and y is the standard deviation of the evaluation metric for the five runs of the model.

| Model | Dice | Accuracy | Precision | Recall | IoU |
|:---:|:---:|:---:|:---:|:---:|:---:|
| (i) | 68.27 (±0.60) | 92.85 (±0.60) | 68.15 (±0.41) | 71.71 (±0.77) | 55.82 (±0.53) |
| (ii) | 71.72 (±0.32) | 94.55 (±0.27) | **78.28 (±0.16)** | 65.45 (±0.39) | 61.08 (±0.49) |
| (iii) | 72.08 (±0.50) | 94.38 (±0.25) | 74.42 (±0.39) | 70.86 (±0.59) | 62.35 (±0.24) |
| (iv) | 71.97 (±0.39) | 94.26 (±0.45) | 73.76 (±0.58) | 69.66 (±0.76) | 62.99 (±0.43) |
| (v) | **74.23 (±0.67)** | **95.88 (±0.42)** | 73.81 (±0.43) | **74.59 (±0.65)** | **65.32 (±0.56)** |

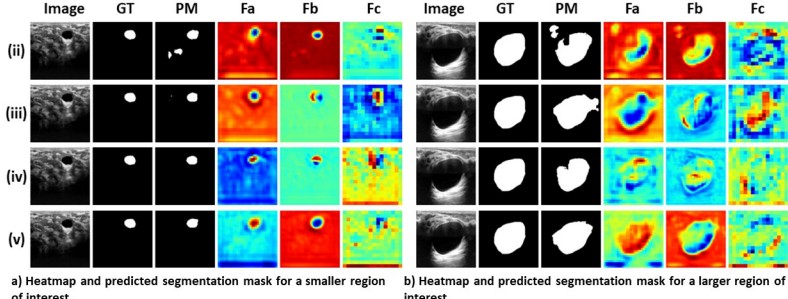

a) Heatmap and predicted segmentation mask for a smaller region of interest.

b) Heatmap and predicted segmentation mask for a larger region of interest.

**Fig 4. Results of the ablation study indicate the improvement in model performance with each experimental modification.** GT and PM are Ground Truth and Predicted Mask, respectively. $F_c$ is the heatmap of the bottleneck layer and it demonstrates the improvement of the model's performance in focusing on the region of interest after the addition of the SWA in the bottleneck layer. $F_a$ and $F_b$ are heatmaps of the features flowing from the first and second encoder layers to the first and second decoder layers via skip connections. It can be seen that $F_a$ and $F_b$ get more enriched with the use of attentions such as CBAM, PAM, and PCBAM.

"The proposed DAU-Net model yields similar results in comparison to the other models considered in the ablation study." To perform this test, we have considered the Mann-Whitney U test [53], a popular non-parametric statistical test. We have compared the Dice and IoU scores from five different runs for each model, as described in the last section (i, ii, iii, and iv) of the base models, with the proposed model (v) to perform this analysis. The results are presented in Table 3. We can safely reject the null hypothesis for each case based on the results provided in Table 3 because the p-value is less than 0.05 (5%) in each case. Furthermore, we have noted that the magnitudes of the results are identical. However, as the Mann-Whitney U test is rank-based and is not dependent on the magnitude of the results, this characteristic does not affect the validity of the statistical test.In conclusion, the statistical analysis using the Mann-Whitney U test provides strong evidence that the proposed DAU-Net model yields statistically significant results compared to the other models considered in the ablation study. This suggests that

**Table 2. Results of the proposed DAU-Net model with 5-fold cross-validation on the BUSI dataset.**

| 5-Fold CV | IoU(%) | DSC(%) |
|---|---|---|
| Fold 1 | 64.85 | 73.58 |
| Fold 2 | 65.43 | 74.14 |
| Fold 3 | 65.17 | 74.29 |
| Fold 4 | 64.79 | 73.21 |
| Fold 5 | 65.91 | 74.61 |
| **Mean** | 65.23 | 73.97 |
| **Std. Dev.** | 0.411 | 0.504 |

**Table 3. Results of the Mann-Whitney U test of the proposed DAU-Net model used for segmenting tumor regions in breast images of the BUSI dataset.**

| Model | p-value (Dice) | p-value (IoU) |
|---|---|---|
| (i) | 0.007 | 0.007 |
| (ii) | 0.007 | 0.007 |
| (iii) | 0.007 | 0.007 |
| (iv) | 0.011 | 0.011 |

**Table 4. Performance metrics of the proposed model with different loss functions.**

| Loss | Accuracy | Dice | Precision | Recall | IoU |
|---|---|---|---|---|---|
| BCE loss | 94.50 | 56.73 | 53.64 | 65.18 | 48.92 |
| Dice loss | 94.69 | 58.94 | 53.01 | 67.79 | 50.31 |
| Focal loss | 94.13 | 53.16 | 76.40 | 49.08 | 43.40 |
| BCE loss + Dice loss | 95.04 | 73.32 | 77.62 | 72.69 | 63.05 |
| BCE loss + Focal loss | 94.97 | 70.61 | 77.48 | 65.63 | 58.27 |
| Dice loss + Focal loss | 94.51 | 69.35 | 72.53 | 70.13 | 54.81 |
| Dice loss + Focal loss + BCE loss | **95.88** | **74.23** | 73.81 | **74.59** | **65.32** |

the use of the dual attention methodology in the present work contributes to the model's effectiveness and reliability.

## Additional experimentation

We have performed a series of experiments utilizing different loss combinations and have consolidated the results in Table 4. This table highlights the ablation study we have conducted on the loss functions employed to train our model. After analyzing Table 4, we have determined the most efficient loss function during model training and selected the optimal model configuration. Our findings reveal that the combination of Dice, BCE, and focal loss yields the highest performance.

## State-of-the-art comparison

We have performed a comparison between our proposed model and several state-of-the-art (SOTA) models along with standard segmentation models. The comparative results, comprehensively evaluating various evaluation metrics, are presented in Tables 5 and 6. The models compared in Table 5 are well-known in the image segmentation field, such as FCN [54], U-Net [6] SegNet [55] and ENC-Net [56]. Our proposed method demonstrates superior performance across the standard models (see Table 5) in terms of Dice score, Precision, and IoU, indicating better overall segmentation accuracy. Furthermore, our proposed model outperforms other advanced models (see Table 6, including ResUNet++ [57], SCAN [58], STAN [59], ColonSegnet [60] and AE-Unet [61], in terms of Dice score and IoU, highlighting its ability to accurately capture the overlap between predicted and ground truth segmentation masks. However, it is important to note that the precision and recall values for our proposed method are slightly lower than some of these models, suggesting a potential trade-off between precision and recall.

To provide specific performance details, our proposed model achieves a Dice score of 74.23, indicating a higher level of similarity between the ground truth and the predicted

**Table 5. Performance comparison with standard segmentation models.** All values are in %. Bold values indicate superior performance.

| Model | Dice | Precision | Recall | IoU |
|---|---|---|---|---|
| FCN [54] | 71.23 | 69.07 | 77.02 | 56.27 |
| UNet [6] | 71.32 | 66.96 | 78.46 | 56.13 |
| SegNet [55] | 72.25 | 68.77 | 80.06 | 60.01 |
| ENC-Net [56] | 72.66 | 68.59 | 79.90 | 57.70 |
| **Proposed** | **74.23** | **73.81** | 74.59 | **65.32** |

**Table 6. Performance comparison with SOTA models.** All values are in %. Bold values indicate superior performance.

| Model | Dice | Precision | Recall | IoU |
|---|---|---|---|---|
| DA-Net [62] | 67.83 | - | 80.38 | - |
| ResUNet++ [57] | 73.85 | **80.10** | 71.43 | 60.02 |
| SK-UNET [63] | 70.90 | - | **80.80** | - |
| SCAN [58] | 72.00 | 73.00 | - | - |
| STAN [59] | 72.00 | 76.00 | - | - |
| ColonSegnet [60] | 73.53 | 76.81 | 76.43 | 62.71 |
| MCF-Net [64] | 71.06 | - | 72.23 | - |
| UNext [65] | 65.94 | - | - | 55.22 |
| AE-Unet [61] | 73.47 | 74.44 | 79.00 | 64.57 |
| RRC-Net [66] | 72.53 | 71.73 | 77.72 | 63.60 |
| MBSNet [67] | 72.81 | - | - | 63.21 |
| U-Net-densenet121 [68] | 73.70 | - | 72.55 | 62.46 |
| **Proposed** | **74.23** | 73.81 | 74.59 | **65.32** |

segmentation masks. Additionally, precision with a value of 73.81 indicates a significant proportion of predicted foreground pixels are indeed correctly identified. The Recall value of 74.59 showcases the model's ability to accurately identify a substantial number of actual foreground pixels. Moreover, the IoU metric, with a value of 65.32, indicates the model's strong capability in accurately delineating regions of interest. Significantly, the proposed method achieves the highest Dice score and IoU out of all the models listed in Table 6, suggesting that it excels in terms of segmentation accuracy and overlap with the ground truth.

Overall, the evaluation results showcase the superior performance of our proposed model compared to the state-of-the-art models. This confirms the effectiveness and robustness of our model in achieving accurate and precise segmentation results, positioning it as a promising solution for various segmentation tasks. However, the slightly lower Precision and Recall value compared to some models may indicate a potential area for improvement. Fig 5 showcases the segmentation results of our proposed model, demonstrating its ability to segment breast tumor regions in ultrasound images accurately. The heatmaps showcase the spatial regions where the SWA and PCBAM layers focus. Furthermore, the heatmap visualization of the proposed model as shown in Fig 4 illustrates the spatial regions where it places more emphasis,

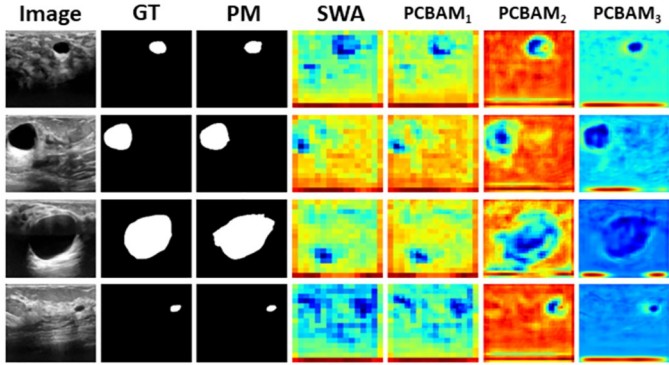

**Fig 5. Results of the proposed segmentation model on images of the BUSI dataset and the heatmaps of SWA and PCBAM layers.** $PCBAM_1$ corresponds to the $PCBAM$ layer just above the SWA layer, $PCBAM_2$ corresponds to the PCBAM layer just above $PCBAM_1$ layer, and $PCBAM_3$ corresponds to the $PCBAM$ layer just above $PCBAM_2$ layer.

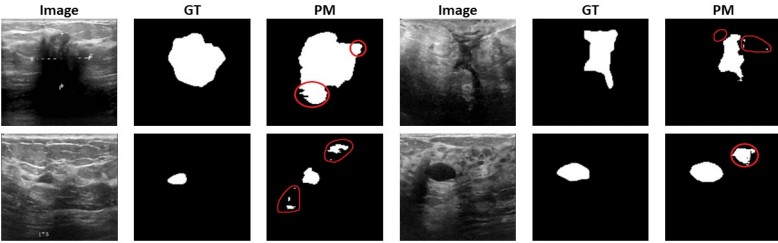

**Fig 6. Illustration of some of the failed cases of our model.** The encircled regions are the misclassified segmented masks. GT and PM represent the Ground Truth and Predicted Mask, respectively.

showing a close resemblance to the ground truth regions for the BUSI dataset. This indicates that the model focuses its attention on relevant areas, contributing to its accurate segmentation performance.

Through these comprehensive evaluations and visualizations, our proposed model showcases its potential to significantly improve breast cancer detection and diagnosis, bringing us closer to the goal of early detection and enhanced patient care.

## Error analysis

Our proposed model has demonstrated excellent performance across various image segmentation tasks, outperforming SOTA models, as depicted in Tables 5 and 6. It is essential to highlight that the precision and recall are relatively lower, indicating instances where non-tumorous regions are misclassified as tumorous and vice-versa. It is important to acknowledge the complexity of the dataset used for evaluation, which presents challenges in achieving perfect segmentation results. Fig 6 illustrates specific cases where our model encounters difficulties, resulting in deviations from the ground truth segmentation. These challenges may arise from dataset complexity, variations in image quality, or the presence of ambiguous features that are hard to accurately delineate.

Despite these challenges, our proposed model demonstrates significant potential and promises a valuable contribution to the field of breast cancer detection. By continuing to address the limitations and exploring further research directions, we aim to enhance the model's segmentation performance and make strides toward more accurate and reliable breast cancer diagnosis.

## Experimentation on the UDIAT dataset

To evaluate the effectiveness of our proposed DAU-Net method, we have conducted assessments on the UDIAT dataset, also known as Dataset B, a well-known collection of breast ultrasound images generously provided by the UDIAT Diagnostic Centre in Sabadell, Spain [69]. This dataset comprises a total of 163 images, consisting of 109 benign and 54 malignant ultrasound images, each accompanied by its respective ground truth mask. The average resolution for both the ultrasound images and the corresponding ground truth masks is $760 \times 570$ pixels. For the evaluation on the UDIAT dataset, we have maintained consistency by using the same set of hyperparameters that are employed in the evaluation of the BUSI dataset. The segmentation results of the proposed method on the UDIAT dataset is shown in Fig 7. Table 7 presents a comprehensive overview of the quantitative performance achieved by our proposed model when compared to previous notable research efforts conducted on this dataset.

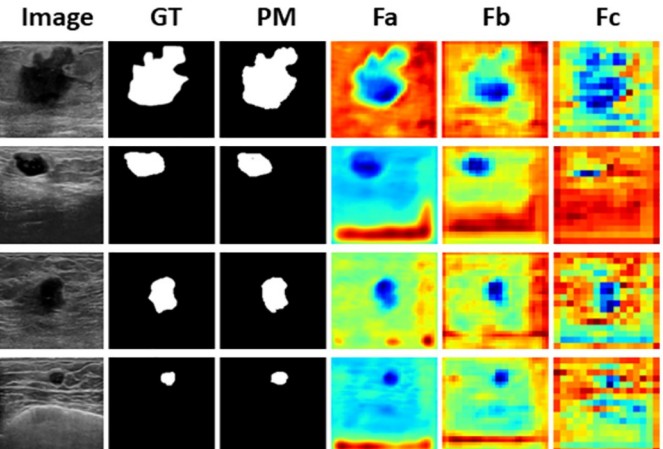

**Fig 7. Predicted mask and heatmap visualization of the proposed model on the UDIAT dataset.** GT and PM represent the Ground Truth and Predicted Mask, respectively. $F_a$, $F_b$, and $F_c$ are the heatmaps of the features flowing from the first and second encoder layers to the first and second decoder layers via skip connections and the bottleneck layer, respectively.

## Conclusion and future scope

In conclusion, breast cancer continues to be a pressing issue worldwide, underscoring the significance of timely identification and precise diagnosis to enhance outcomes. With the latest strides in deep learning, CAD solutions have emerged as promising tools in this domain. The current study introduces a novel segmentation technique called the PCBAM attention-based Residual U-Net model for detecting breast tumors in ultrasound images. Our approach has delivered satisfactory outcomes, revealing its potential to improve breast cancer detection and diagnosis. Nonetheless, it is crucial to recognize the constraints of our proposed method.

During the error analysis section, several potential areas for future research were identified. One particularly promising direction involves investigating multi-modal architectures that integrate multiple forms of data sources within the model. Such an approach has the potential to improve performance and deepen our comprehension of complex breast cancer detection challenges by leveraging the complementary insights provided by various modalities. Another avenue for future research is to enhance datasets through techniques such as data synthesis or generation, which can increase their diversity and size, thereby enhancing generalization and robustness.

**Table 7. Performance comparison of the proposed model with past methods on UDIAT dataset.** All values are in %. Bold values indicate superior performance.

| Model | Dice | Precision | IoU |
|---|---|---|---|
| SegNet [55] | 70.80 | 85.00 | 60.00 |
| CE-Net [70] | 72.00 | 74.00 | 61.00 |
| MultiResUNet [71] | 75.00 | 79.00 | 66.00 |
| SCAN [58] | 74.00 | 75.00 | 65.00 |
| U-Net [6] | 75.00 | 78.00 | 65.00 |
| STAN [59] | 78.20 | 80.00 | - |
| **Proposed** | **78.58** | **85.85** | 64.71 |

Additionally, evaluating the robustness of the model could provide valuable insights. Performing segmentation on other types of medical images, beyond breast ultrasound images, would be beneficial in assessing the model's versatility and applicability in diverse medical imaging tasks. In conclusion, our proposed PCBAM attention-based Residual U-Net model shows promise in breast tumor detection in ultrasound images. However, continued research in multi-modal architectures, dataset augmentation, and evaluation of other medical images will contribute to the advancement of accurate and reliable breast cancer diagnosis.

## Acknowledgments

We are thankful to the Center for Microprocessor Applications for Training Education and Research (CMATER) research laboratory of the Computer Science and Engineering Department, Jadavpur University, Kolkata, India, for providing infrastructural support to this research project.

## Author Contributions

**Investigation:** Payel Pramanik, Ayush Roy, Erik Cuevas, Ram Sarkar.

**Writing – original draft:** Payel Pramanik, Ayush Roy, Erik Cuevas, Ram Sarkar.

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
