## [Decision Letter · Decision Letter 0]

21 Dec 2023

PONE-D-23-35838DAU-Net: Dual Attention-aided U-Net for Segmenting Tumor in Breast Ultrasound ImagesPLOS ONE

Dear Dr. Perez-Cisneros,

Thank you for submitting your manuscript to PLOS ONE. After careful consideration, we feel that it has merit but does not fully meet PLOS ONE’s publication criteria as it currently stands. Therefore, we invite you to submit a revised version of the manuscript that addresses the points raised during the review process.

We look forward to receiving your revised manuscript.

Kind regards,

Chenchu Xu, Ph.D

Academic Editor

PLOS ONE

Journal Requirements:

3. We note that your Data Availability Statement is currently as follows: All relevant data are within the manuscript and its Supporting Information files

Additional Editor Comments:

The reviewers suggest improvements in several key areas: expanding dataset validation for better generalizability, enhancing language clarity, providing more detailed descriptions of innovations like PCBAM and SWA, clarifying statistical methods and results, and updating the literature review to include the latest research. Additionally, the manuscript should address potential overfitting as indicated in Fig. 4, present comparative predictive results, apply k-fold cross-validation for fairness, and consider the implications of image resizing dimensions. These revisions are essential for enhancing the manuscript's overall quality and relevance to the field.

Reviewers' comments:

Reviewer's Responses to Questions

**Comments to the Author**

1. Is the manuscript technically sound, and do the data support the conclusions?

Reviewer #1: Yes

Reviewer #2: Partly

2. Has the statistical analysis been performed appropriately and rigorously? 

Reviewer #1: Yes

Reviewer #2: Yes

3. Have the authors made all data underlying the findings in their manuscript fully available?

Reviewer #1: Yes

Reviewer #2: Yes

4. Is the manuscript presented in an intelligible fashion and written in standard English?

Reviewer #1: Yes

Reviewer #2: Yes

5. Review Comments to the Author

Reviewer #1: In my opinion, some important problems in this paper need to be thoroughly revised:

1） The proposed method is only verified on one kind of dataset, which I think is very insufficient. I hope the authors can experiment with other datasets.

2） The English expression of this paper is not very standardized, which brings some confusion to the reader's understanding. I think it could be polished by someone more professional.

3） The proposed PCBAM and SWA are the main innovations of this paper. Here, PCBAM is already a relatively conventional method, so it can be considered meaningful. However, the SWA module authors did not describe it accurately enough in to “Shifted Window Attention” section. I did not understand the specifics of the module. Moreover, the authors can refer to these papers: https://doi.org/10.1016/j.media.2023.102980, https://doi.org/10.1016/j.compmedimag.2022.102054, etc.

4） What does Fig. 4 mean? Prove that the model is overfitted. If the results on the validation set are so volatile, are the results of the test set still reliable?

5） Table 2 shows the "Mann-Whitney U test" or "p-value test". I think these are two different evaluation methods. Moreover, are all the p-values of the 4 ablation methods shown in the table and the proposed DAU-Net model 0.0286? I remain skeptical.

6） Is there a difference between “mlp“ in formula (1) and ”DL“ in formula (2)?

Reviewer #2: Breast lesion segmentation is a very interesting work. In this paper, they present a novel Dual Attention-aided U-Net (DAU-Net) for breast ultrasound segmentation. Although the study is interesting, this paper has a few concerns for publication:

1. In Abstract, it would have been interesting to quantify the results obtained at this level, to guide readers on the performance of your model.

2. The authors have omitted much of the latest research work. Especially the work of 2022 and 2023 is missing. It is recommended that the authors add the latest literature to help readers gain more comprehensive knowledge. The following work can be referred:

[1] https://doi.org/10.1109/TMI.2022.3226268

[2] https://doi.org/10.1016/j.patcog.2023.109728

[3] https://doi.org/10.1016/j.cmpb.2022.107086

3. In Fig. 6, the author needs to display the prediction results of each method.

4. To ensure the fairness of the experiment, k-fold cross validation is necessary. It can be referred to in future research.

5. Would resizing the original image to 128 be too small?

6. PLOS authors have the option to publish the peer review history of their article (what does this mean?). If published, this will include your full peer review and any attached files.

Reviewer #1: No

Reviewer #2: No

---

## [Author Response · Author response to Decision Letter 0]

29 Jan 2024

We are very grateful to the Editor for considering our manuscript for review and we are very

thankful to the reviewers for providing us with valuable suggestions and giving feedback about

our manuscript. We have made the required changes to address the editor’s and reviewers’

comments. Please note that the changes are highlighted in the revised manuscript and herein as

blue for Reviewer #1 and green for Reviewer #2 for the convenience of the reviewers. Replies to

each of the reviewers’ comments are also listed below

---

## [Decision Letter · Decision Letter 1]

29 Feb 2024

PONE-D-23-35838R1DAU-Net: Dual Attention-aided U-Net for Segmenting Tumor in Breast Ultrasound ImagesPLOS ONE

Dear Dr. Perez-Cisneros,

Thank you for submitting your manuscript to PLOS ONE. After careful consideration, we feel that it has merit but does not fully meet PLOS ONE’s publication criteria as it currently stands. Therefore, we invite you to submit a revised version of the manuscript that addresses the points raised during the review process.

We look forward to receiving your revised manuscript.

Kind regards,

Chenchu Xu, Ph.D

Academic Editor

PLOS ONE

Journal Requirements:

Reviewers' comments:

Reviewer's Responses to Questions

**Comments to the Author**

1. If the authors have adequately addressed your comments raised in a previous round of review and you feel that this manuscript is now acceptable for publication, you may indicate that here to bypass the “Comments to the Author” section, enter your conflict of interest statement in the “Confidential to Editor” section, and submit your "Accept" recommendation.

Reviewer #1: (No Response)

Reviewer #2: All comments have been addressed

2. Is the manuscript technically sound, and do the data support the conclusions?

Reviewer #1: Yes

Reviewer #2: Yes

3. Has the statistical analysis been performed appropriately and rigorously? 

Reviewer #1: Yes

Reviewer #2: Yes

4. Have the authors made all data underlying the findings in their manuscript fully available?

Reviewer #1: Yes

Reviewer #2: No

5. Is the manuscript presented in an intelligible fashion and written in standard English?

Reviewer #1: Yes

Reviewer #2: Yes

6. Review Comments to the Author

Reviewer #1: Is it true that the training losses and accuracy in Fig. 4 are stable, but the validation losses and accuracy are so volatile? This is a problem that I don't think has been effectively addressed.

Reviewer #2: Thanks for the effort put in as a solution to my doubt. The revised manuscript may be considered for acceptance.

7. PLOS authors have the option to publish the peer review history of their article (what does this mean?). If published, this will include your full peer review and any attached files.

Reviewer #1: No

Reviewer #2: No

---

## [Author Response · Author response to Decision Letter 1]

30 Mar 2024

We are very grateful to the Editor for considering our manuscript for review and we are very

thankful to the reviewers for providing us with valuable suggestions and giving feedback about

our manuscript. We have made the required changes to address the editor’s and reviewers’

comments.

---

## [Decision Letter · Decision Letter 2]

30 Apr 2024

DAU-Net: Dual Attention-aided U-Net for Segmenting Tumor in Breast Ultrasound Images

PONE-D-23-35838R2

Dear Dr. Perez-Cisneros,

We’re pleased to inform you that your manuscript has been judged scientifically suitable for publication and will be formally accepted for publication once it meets all outstanding technical requirements.

Kind regards,

Chenchu Xu, Ph.D

Academic Editor

PLOS ONE

Additional Editor Comments (optional):

Reviewers' comments:

Reviewer's Responses to Questions

**Comments to the Author**

1. If the authors have adequately addressed your comments raised in a previous round of review and you feel that this manuscript is now acceptable for publication, you may indicate that here to bypass the “Comments to the Author” section, enter your conflict of interest statement in the “Confidential to Editor” section, and submit your "Accept" recommendation.

Reviewer #1: All comments have been addressed

Reviewer #2: All comments have been addressed

2. Is the manuscript technically sound, and do the data support the conclusions?

Reviewer #1: Yes

Reviewer #2: Yes

3. Has the statistical analysis been performed appropriately and rigorously? 

Reviewer #1: Yes

Reviewer #2: Yes

4. Have the authors made all data underlying the findings in their manuscript fully available?

Reviewer #1: No

Reviewer #2: Yes

5. Is the manuscript presented in an intelligible fashion and written in standard English?

Reviewer #1: Yes

Reviewer #2: Yes

6. Review Comments to the Author

Reviewer #1: The questions I raised have been revised, and the author has made a detailed reply. I think this article is acceptable.

Reviewer #2: (No Response)

7. PLOS authors have the option to publish the peer review history of their article (what does this mean?). If published, this will include your full peer review and any attached files.

Reviewer #1: No

Reviewer #2: No

---

## [Editor Report · Acceptance letter]

21 May 2024

PONE-D-23-35838R2 

PLOS ONE

Dear Dr. Perez-Cisneros, 

I'm pleased to inform you that your manuscript has been deemed suitable for publication in PLOS ONE. Congratulations! Your manuscript is now being handed over to our production team.

Kind regards, 

on behalf of

Dr. Chenchu Xu 

Academic Editor

PLOS ONE